# Low Replicative Stress Triggers Cell-Type Specific Inheritable Advanced Replication Timing

**DOI:** 10.3390/ijms22094959

**Published:** 2021-05-07

**Authors:** Lilas Courtot, Elodie Bournique, Chrystelle Maric, Laure Guitton-Sert, Miguel Madrid-Mencía, Vera Pancaldi, Jean-Charles Cadoret, Jean-Sébastien Hoffmann, Valérie Bergoglio

**Affiliations:** 1Centre de Recherches en Cancérologie de Toulouse (CRCT), UMR1037 Inserm, University Paul Sabatier III, ERL5294 CNRS, 2 Avenue Hubert Curien, 31037 Toulouse, France; lilas.courtot@inserm.fr (L.C.); ebourniq@uci.edu (E.B.); guittonsert@gmail.com (L.G.-S.); miguel.madrid-mencia@inserm.fr (M.M.-M.); vera.pancaldi@inserm.fr (V.P.); 2Université de Paris, CNRS, Institut Jacques Monod, DNA Replication Pathologies Team, F-75006 Paris, France; chrystelle.maric@ijm.fr; 3Barcelona Supercomputing Center, 08034 Barcelona, Spain; 4Laboratoire de pathologie, Laboratoire d’excellence Toulouse Cancer, Institut Universitaire du Cancer-Toulouse, Oncopole, 1 Avenue Irène-Joliot-Curie, CEDEX, 31059 Toulouse, France

**Keywords:** DNA replication stress, DNA replication timing, chromatin accessibility, DNA damage

## Abstract

DNA replication timing (RT), reflecting the temporal order of origin activation, is known as a robust and conserved cell-type specific process. Upon low replication stress, the slowing of replication forks induces well-documented RT delays associated to genetic instability, but it can also generate RT advances that are still uncharacterized. In order to characterize these advanced initiation events, we monitored the whole genome RT from six independent human cell lines treated with low doses of aphidicolin. We report that RT advances are cell-type-specific and involve large heterochromatin domains. Importantly, we found that some major late to early RT advances can be inherited by the unstressed next-cellular generation, which is a unique process that correlates with enhanced chromatin accessibility, as well as modified replication origin landscape and gene expression in daughter cells. Collectively, this work highlights how low replication stress may impact cellular identity by RT advances events at a subset of chromosomal domains.

## 1. Introduction

DNA replication is a highly complex process that ensures the accurate duplication of the genome, hence the faithful transmission of genetic material to the cell progeny. DNA replication occurs during S phase through replisome activity, but it requires important upstream regulation during the G1 phase and checkpoints in G2 phase, together with a tight control throughout the process itself. Multicomplex replication machinery performs the coordinated initiation of DNA synthesis at hundreds of replication origins spread throughout the whole length of the genome [1]. Adjacent origins that initiate DNA replication at the same time have been called “replicon clusters” [2], giving rise to chromosomal domains replicating synchronously at given times during the S phase. This coordination of the temporal program of DNA replication, called “replication timing” (RT), allows a complete and faithful duplication of the entire genome before cell division.

The RT program is modified during organism development and cell differentiation [3,4] and is coupled with gene expression, chromatin epigenome, and nuclear 3D compartmentalization [5,6,7,8]. In somatic cells, the RT pattern is very robust through cell generations [8,9,10,11] with early-replicating DNA residing deep within the nucleus that correspond to the A compartment containing active chromatin. The later-replicating regions occur at the nuclear periphery or near the nucleolus [9,12,13] that correspond to the B compartment containing inactive chromatin. Additional complex associations have been highlighted such as the link between early-replicating regions and GC nucleotides enrichment, enhanced gene expression, and active epigenetic marks corresponding to open or euchromatin. Conversely, late-replicating regions tend to be enriched in AT nucleotides, show low gene content, and have heterochromatin repressive epigenetic marks [12,13].

DNA replication stress is defined as the slowing or stalling of the replication fork, resulting in inefficient DNA replication. Many exogenous or endogenous sources of impediment on DNA, as well as pathological perturbations such as oncogene activation, conflicts between DNA replication and transcription, or shortage of nucleotides affect the progression of replication forks, inducing replication stress [14,15,16,17,18]. Experimentally, replication stress can be induced by the specific inhibition of replicative DNA polymerases by treatment with the drug aphidicolin. Notably, low doses of aphidicolin (0.1 to 0.6 µM) are well known to cause the induction of common fragile sites (CFS) expression and the generation of under-replicated DNA that leads to DNA damage transmission [19,20,21,22,23]. CFS are chromosomal regions harboring cancer-related genes [24] that are prone to breakage upon replication stress [25] and whose instability is often observed at the early stages of carcinogenesis [26]. The fragility of these chromosomal regions has been widely studied, revealing incomplete DNA replication before mitosis mainly due to conflicts with large transcription units [27,28,29] and/or origin paucity [30,31]. 

Evidence of aberrant RT in many genetic diseases and cancers suggests that this cellular process is important for genomic stability [32,33,34]. Interestingly, replication stress inducing CFS expression also affects the RT of these specific chromosomal domains [27,29]. The extent to which these RT changes influence the tumor transformation process is still largely unknown. 

The major aim of this study was to explore whether low replication stress differentially affects RT regarding to the cell type. To do so, we characterized and compared the impact of mild replication stress induced by low doses of aphidicolin on four cancer and two non-cancer human cell lines (colon, blood, osteoblast, retina, and lung) (Appendix A). Our experiments revealed that a low dose of aphidicolin displays a quantitative and qualitative cell-type-specific impact on the RT, promoting RT delays and RT advances. We demonstrated that the two cancer cell lines RKO and K562 share strong RT advances that are poorly enriched in DNA damage signaling histone marks, while being characterized by increases in chromatin accessibility in response to aphidicolin. Finally, in RKO colon cancer cells, we observed that the persistence of RT advances in daughter cells released from replication stress is correlated to modification of chromatin loop size and pre-replication complex (pre-RC) loading in G1. Altogether, our results indicate that low replication stress leads to cell type-specific RT modifications and reveal that strong RT advances characterize cells with higher chromatin flexibility through the modification of the DNA replication program and expression of specific cancer-related genes. 

## 2. Results

### 2.1. Low Replication Stress Induces Cell-Type Replication Timing Changes

In this study, we used six well-characterized cell lines that are common models (HCT116, RKO, U2OS, K562, MRC5-N, and RPE-1) and differ in tissue origin, tumorigenicity, differentiation stage, and molecular characteristics such as oncogene expression, genetic instability type, and telomere maintenance mechanisms (Appendix A). In order to evaluate cellular responses to mild replication stress, we treated cells with 0.2 µM aphidicolin and DMSO as a control. Aphidicolin treatment induces a low level of Chk1 phosphorylation on serine 345 compared to acute HU treatment, and we noticed that MRC5-N and RPE-1 cells have the lowest P-Chk1 induction (Appendix A). The duration of treatment was adapted to each cell line in order to treat a maximum of S-phase cells for a single generation (Appendix A). For this purpose, BrdU has been incorporated during APH treatment in order to follow S-phase treated cells in the subsequent cell cycle phases by cytometry (Appendix A). To be assured that cells were treated during one S-phase only, the APH treatment duration was adapted so only few BrdU-positive cells reach G1-phase (Appendix A). This treatment induces an accumulation of cells in S-phase without cell cycle arrest, allowing us to perform the RT experiments (Appendix A). 

To study RT under low replication stress, BrdU was added to the culture medium before cell sorting into Early (S1) and Late S-phase (S2) fractions, and the neo-synthesized DNA was hybridized on human whole genome microarrays, as previously described [35,36,37]. RT differential analyses were performed on biological replicate experiments using the START-R suite software [38] and only significant modifications between aphidicolin and control condition were retained (Figure 1A and Appendix A). As mentioned by Hadjadj et al., the START-R suite detects constant timing regions (CTR) but also it identifies temporal transition regions (TTR) and allows robust statistical analysis to detect differences between two experiments. For these experiments, after normalization done by START-R, we apply the mean method with a *p*-value < 0.05 to significantly detect the disturbs in the replication timing between two conditions.

We calculated the genome-wide percentage of altered RT in the different cell lines in response to aphidicolin treatment (Figure 1B). We first noticed that normal RPE-1 and MRC5-N cells were the least impacted with respectively 1.54% and 2.01% of the genome undergoing RT alterations, while the RT in the four cancerous cells was globally more affected by low replication stress (3.09–6.62%). Analysis of the aphidicolin-RT-impacted loci (aRTIL) led to the identification of regions with significant RT delays (DEL aRTIL) and RT advances (ADV aRTIL). As previously reported in immortalized human foreskin fibroblasts (BJ-hTERT) and immortalized human B lymphocytes [27,29] we found that DEL aRTIL represent the majority (77%) of impacted loci in hTERT-immortalized retinal pigment epithelial cell line (RPE-1 hTERT) but also in the colon cancer cell line HCT116 cells (Figure 1C). An equal proportion between DEL and ADV aRTIL was observed in the embryonic lung cells MRC5-N (Figure 1C). Interestingly, in the osteosarcoma U2OS cells, in the K562 leukemic cells, and especially in colon cancer RKO cells, a higher proportion of ADV aRTIL was identified (Figure 1C). We noticed a considerable shift from late to early S-phase in large genomic regions of RKO cells and, to a lesser extent, in K562 cell lines (Figure 1D and Appendix A). To confirm that RT advances were not experimental artefacts, we validated three ADV aRTIL by BrdU-ChIP-qPCR approach in the RKO cell line, confirming that the identified advanced domains are replicated significantly earlier (relative to BrdU incorporation) upon aphidicolin treatment (Appendix A). We also confirmed that unmodified early and late RT domains were mostly replicated in their respective early-S and late-S fractions, without significant difference between untreated and aphidicolin-treated conditions (Appendix A). Overall, this whole genome RT differential analysis indicates that depending on the cell type, RT is differentially impacted by replication stress in both a quantitative and qualitative manner. 

#### 2.1.1. Genomic and Epigenomic Characterization of aRTIL

We first noticed that ADV aRTIL are normally replicated later than DEL aRTIL in all six cell lines and that ADV aRTIL from RKO and K562 are replicated even later than those of other lineages (Figure 2A). Furthermore, analysis of other genomic parameters such as aRTIL size (Figure 2B), GC content (Figure 2C), origin coverage (Appendix A), and gene coverage (Appendix A) show that RKO and K562 advanced domain share genomic features, which are distinct from ADV aRTIL of the other cell lines and from DEL aRTIL. Using available ENCODE ChIP-seq data in K562 and HCT116 (Appendix A), we further demonstrated that epigenomic features of ADV aRTIL of K562 are closely related to late replicated domains while ADV aRTIL from HCT116 are characterized by mid S-phase histone mark patterns (Figure 2D and Appendix A). The singularity of ADV aRTIL features found in RKO and K562 led us to analyze if these domains could be shared by these two cell lines. Overlaps of ADV and DEL aRTIL in the six cell lines were compared to randomized genomic regions using Jaccard index, which is a statistic used for gauging the similarity and diversity of sample sets. The results presented in Figure 2E top and Appendix A show that DEL aRTIL are non-randomly shared between all the cell lines, excepted the K562, in which DEL aRTIL seem to be more cell type specific. For ADV aRTIL we identified a significant overlap between RKO and K562 ADV aRTIL compared to random (Figure 2E bottom and Appendix A). To a lesser extent, we also noticed significant overlap between MRC5-N and U2OS ADV aRTIL, while in HCT116 and RPE-1 cell lines, these latest are more cell type specific (Figure 2E bottom).

#### 2.1.2. ADV aRTIL Are Related to CFS but Not Enriched in DNA Damage Signaling Histone Marks

Given that CFS are the most sensitive chromosomal regions to replication stress and that RT delays have been described in these fragile loci, we wondered if aRTIL overlap with CFS. To answer this question, we analyzed the intersection between aRTIL and 59 CFS mapped by conventional cytogenetics in several cell types [27]. First, we found 22 to 44% of the 59 CFS analyzed harboring RT alterations under aphidicolin treatment (Figure 3A). Non-random overlap of DEL aRTIL and ADV aRTIL with CFS was investigated. Quite surprisingly, the most important overlap with CFS was observed for RKO and K562 ADV aRTIL (Figure 3B), while overlaps between DEL aRTIL and CFS are comparable to random (Appendix A). This result indicates that even if some DEL aRTIL intersect with CFS, there is no significant enrichment. Nevertheless, when we quantified large genes (>400 kb) coverage, that were generally associated to CFS [27,28,29,31,39], we observed that large genes are exclusively associated to DEL aRTIL in HCT116, MRC5-N, and RPE-1 cells (Appendix A), while they are also identified in ADV aRTIL from RKO, K562, and U20S cells (Appendix A). In addition, we cannot exclude the possibility that we missed some delays in the CFS. Indeed, the late S phase fraction (S2) does not cover the G2 phase, so we did not analyze cells with under-replicated CFS, which have been described to have a delayed RT in response to aphidicolin [30]. Overall, our results do not allow us to firmly conclude on the association of DEL aRTIL with CFS, but we have demonstrated that CFS can harbor RT advances in RKO and K562 cell lines. 

It has been recently reported that macroH2A1.2, a variant from the canonical H2A [40], having roles both in replication stress response and in cell fate decisions [41,42,43,44], is more abundant at recurrent fragile sites targeted by γ-H2AX in response to aphidicolin treatment [45]. Using available ChIP-seq data, we analyzed the coverage of histone variants γ-H2AX and mH2A1.2 in K562 with or without aphidicolin treatment (Appendix A). As we did for epigenomic marks, we compared DEL aRTIL coverage with early, mid, and late S-phase regions. We confirm that without aphidicolin, early-S regions are the most enriched in γ-H2AX and mH2A1.2 histone variants (Figure 3C,D) [46]. As expected, aphidicolin induces a significant increase in γ-H2AX and mH2A1.2 in most genomic regions. Interestingly, aphidicolin treatment does not modify the coverage of γ-H2AX and mH2A1.2 within ADV aRTIL (Figure 3C,D and Appendix A). This suggests that in K562 RT, advances could be protective against DNA damages. 

Collectively, these data demonstrate that the level and the type (ADV or DEL) of RT modifications in response to mild replication stress depend on the cell line. We revealed a specific ADV aRTIL signature in response to replication stress in RKO and K562 cell lines, which is characterized by late replicating genomic and epigenomic features, CFS enrichment, and no scars of DNA damage under aphidicolin treatment.

### 2.2. ADV aRTIL Can Be Transmitted to Daughter Cells

It was described that RT is a robust process maintained during successive cell cycles in order to ensure cell identity. RT changes have been observed during development in association with gene expression modification relative to cellular differentiation [3,9,10].

We first wondered if RT changes under replication stress in mother cells could be preserved beyond cell division and transmitted to daughter cells. To address this question, we released the six cell lines at the end of the treatment with aphidicolin or DMSO for a determined period of time in each cell line (Appendix A). To ensure that the RT analysis is strictly derived from unstressed S-phase daughter cells (N + 1), we monitored that mother-treated cells, identified by EdU incorporation at the end of treatment, exit the S-phase during the release and that a maximum of daughter cells reached the subsequent S-phase after the release (Appendix A). DNA replication timing results indicate that DEL and ADV aRTIL observed in mother cells were no longer detected in the next cell generation of K562, HCT116, U2OS, MRC5-N, and RPE1 (Figure 4A). Strikingly, in RKO cells, while DEL aRTIL returned to normal RT, the majority (28 of 49, 57%) of the strongest and largest ADV aRTIL detected in T0 mother cells were transmitted to the next cell generation, albeit with lower amplitude (Figure 4B and Appendix A). As in mother cells, these replication timing advances were validated by BrdU-ChIP-qPCR (Appendix A). These results indicate that while the majority of the RT modifications within aRTIL are reversible and tend to be eliminated through mitosis, severe ADV aRTIL can be transmitted to the next generation. 

It has been established that RT switches in human cells can be linked to developmental genes expression [3,12,47,48]. Nonetheless, the exact correlation between RT and gene expression is not entirely clear. Indeed, several studies have discovered genomic sequences that do not fit the general correlation between gene expression and RT [49,50,51]. It has been shown beforehand that low replicative stress, comparable to the one we used, has no major impact on the global gene expression [29,52]. However, we checked if in our experimental conditions, aphidicolin would induce specific gene expression regulation during treatment in mother cells and after release in daughter cells. We performed gene-expression profiling by microarray in RKO and RPE-1 cells using the same conditions as for RT analysis. In mother cells, we found that aphidicolin treatment significantly impacts the expression of 14 genes in RKO (Figure 4C) that are mainly associated to a direct stress response induced by aphidicolin (Appendix A) and seven genes in RPE-1 (Figure 4D). In RKO daughter cells released from the stress, we observed that 20 genes, different from those found at T0, have a significant different level of expression compared to untreated cells (Figure 4E,F), while in RPE-1 cells, the RT and transcription program are not anymore impacted in the daughter cells (Figure 4E). Interestingly, 16 out of 20 genes whose expression are altered in RKO daughter cells have already been described to be associated to cancer development, aggressiveness, and poor prognosis (Appendix A). The genes impacted in terms of their expression at T0 and N + 1 are not identical and none of these transcriptionally affected genes fell inside the RKO transmitted ADV aRTIL, which indicates that the transmission of RT and gene expression changes are two distinct consequences of replication stress. 

#### 2.2.1. Aphidicolin Affects the RT-Related Identity of Cells

It has been reported that the RT profile of somatic cells is closely related to the cell type and tissue origin [9,10]. A hierarchical clustering based on RT values in the different cell lines showed that in absence of replication stress, non-cancer cells clustered together (cluster 1) and were separated from cancer cells (cluster 2) (Figure 4G). In cluster 2, we noticed that the replication timing of RKO is closer to HCT116, which is consistent with the fact that they are both colon cancer cell lines with microsatellite instability due to mismatch repair deficiency (MMR-) (Appendix A). Upon aphidicolin treatment, the two distinct clusters of cancer and non-tumor cells remain separated. However, inside cluster 2, the relatedness between cancer cells is altered. Indeed, the RT of RKO cells appears to be closer to that of K562 cells (Figure 4H). Overall, this observation indicates firstly that without any replication stress, RT itself can discriminate non-tumor from cancer cells as previously described [53], and secondly, that aphidicolin treatment affects the RT-related identity of RKO cells. 

#### 2.2.2. Aphidicolin Modulates Chromatin Accessibility within RKO ADV aRTIL

Since replication timing and transcription activity are both regulated by chromatin structure, and given that we independently identified ADV aRTIL and modification of expression of specific genes in RKO cells, we decided to analyze the impact of this low replicative stress on chromatin structure. 

We performed the Assay for Transposase Accessible Chromatin with high-throughput sequencing (ATAC-seq), which is a method for measuring chromatin accessibility genome-wide [54,55]. In three independent experiments, we observed a notable increase in the number of ATAC-seq peaks identified in the aphidicolin condition (Figure 5A). In addition, the average peak value analysis that considers the peak value and coverage at different genomic regions reveals a genome-wide effect of aphidicolin treatment on chromatin accessibility (Figure 5B,C). ATAC seq peaks enrichment was not significantly impacted at the whole genome promoters and gene bodies (Appendix A). We did not observe significant chromatin accessibility modification for genes within RKO ADV aRTIL (Appendix A). This result further supports our gene expression data, as only transcriptional changes at specific gene outside aRTIL were observed under aphidicolin treatment in RKO cells (Figure 4D). However, confronting RT and ATAC-seq data, we identified a specific increase of ATAC-seq peaks value within ADV aRTIL and not within DEL aRTIL, early, or late replicating regions (Figure 5B,C and Appendix A). This result demonstrates that chromatin remodeling induced by aphidicolin occurs within ADV aRTIL intergenic regions independently of the genomic and epigenomic determinants defining early and late RT. 

RT is faithfully established at the beginning of the G1-phase in each cell cycle, at a precise time named the “timing decision point” or TDP [56,57]. In G1, the RT program setting up is dependent on 3D nuclear replication domains’ organization through chromatin loop formation mediated by the Rif1 protein [58]. Moreover, at the G1/S transition, chromatin loops are also maintained by the transient recruitment of pre-replication complex proteins and active origins to the nuclear matrix (NM) [59,60,61,62]. 

To investigate if the persistence of ADV aRTIL in RKO daughter cells could be linked to changes in the chromatin loop organization in G1-phase, we performed fluorescent DNA halo experiments to evaluate the chromatin loops size in G1/S. We used RKO cells as positive control and RPE-1 cells as negative control. We measured the maximum fluorescence halo radius (MFHR) formed around the nuclear matrix (NM) and noticed a significant shrinkage of DNA loops in the aphidicolin-released RKO cells (Figure 5D,E), while no effect was measured in RPE-1 cells (Appendix A). To test if this reduced halo size correlates with an increase in licensed origins, we quantified the loading of pre-replication complex components onto the chromatin under the same conditions. Our results clearly show an increase of MCM2 and p-MCM2 loading onto the chromatin in G1/S of RKO aphidicolin-released daughter cells (Figure 5F), while this was not the case in RPE-1 cells (Appendix A). Therefore, the transmission of ADV aRTIL in RKO daughter cells is associated to a decrease in chromatin loop size and an increase in pre-RC proteins loading in G1 phase, predicting greater activation of replication origin in the next S phase.

These results indicate increased chromatin flexibility of RKO cells in response to low replication stress that is correlated to replication program alterations, changes in replicon organization, and modification of specific genes expression in the next cell generation. 

## 3. Discussion

In the present work, we demonstrated that the replication timing program is affected in response to low dose of aphidicolin. In all studied cell lines, RT alterations are qualitatively and quantitatively dependent on the cell type (Figure 1). Interestingly, we demonstrated that normal cells (MRC5-N and RPE-1) are quantitatively less affected by RT modifications than cancer cells. This result could be related to the very low level of P-Chk1 we detected for these cell types in response to aphidicolin treatment (Appendix A). Among cancer cells, RT modifications in response to low replication stress are more heterogeneous. Notably, we observed that RKO and K562 cells lines have a particular aRTIL signature characterized by major ADV aRTIL positioned on large domains. These regions are late replicated and poor in GC, genes, and constitutive replication origins contents (Figure 2). Analyses of RT in daughter cells released from the aphidicolin drug show that the transmission of RT modifications was restricted to the strong ADV aRTIL identified in the RKO cell line. None of the K562 ADV aRTIL were transmitted to the daughter cells (Figure 4). The main difference might lie in ADV aRTIL amplitude differences between the two cell lines. In RKO cells, ADV aRTIL correspond to sharp shifts from late to early RT, while in K562, the shift is less pronounced, leading to late to mid/late RT. As RT is related to A/B compartmentalization in such a way that early regions are located in the A compartment and late ones are located in the B compartment [32,63,64], we suggest that the transmission of RT modifications implies a B to A compartment transition of the affected regions.

The nuclear lamina, a meshwork of A- and B-type lamins, participates in the mechanical properties of the nucleus, chromosome architecture, and chromatin dynamics [65,66]. It was recently shown that the lamina controls the RT of late domains called Lamin-Associated Domains [67]. Furthermore, loci involved in late to early RT transitions undergo detachment from the lamina, leading to compartment changes [68], and it was shown that loss of lamin A function increases chromatin dynamics in the nuclear interior [69,70]. Interestingly, we determined that lamin A/C expression is totally abolished at the mRNA and protein levels in RKO cells compared to other cell lines (Appendix A). This observation indicates that chromatin organization is more flexible in RKO cells and could explain why they are more permissive to late to early replication timing transition in response to replication stress. Indeed, we can suggest that ADV aRTIL are already poorly interacting with the nuclear lamina in RKO cells and that aphidicolin treatment completely abolishes this interaction, leading to B to A compartment change. 

The general correlation between RT and gene expression has been the subject of study for a long time with, however, many conflicting results [3,8,51,68,71,72,73,74,75,76,77]. In the present work, we could not directly correlate RT advances with the upregulation of gene expression in response to low replication stress (Figure 4). In our conditions, the aphidicolin treatment is restricted to a single S-phase, and we observed that late to early ADV aRTIL are transient, since their transmission to the daughter cells leads to a decrease in their amplitude. Thus, we suggest that the increase in transcription induced by replication timing advances requires the maintenance of late to early timing changes for more than one cell cycle. It would be interesting to investigate further this point following chronic treatment with a low dose of aphidicolin. Even if RT advances are not directly related to changes in the expression of the genes contained inside aRTIL, we observed that the transmission of ADV aRTIL in RKO is concomitant to significant changes in the expression of specific cancer-related genes in daughter cells. This result suggests that the transmission of RT modifications and gene expression changes can be both induced by the chromatin organization changes after aphidicolin treatment (Figure 5). Thus, we demonstrated that RKO cells display a high flexibility in the chromatin organization, leading to stronger modifications of the replication program, replication origin usage, and gene expression in response to replication stress (Figure 6).

It has been described that the activation of replication stress induces dormant origins within a given S-phase [78,79,80,81,82,83], leading to DNA loop size reduction in the next G1-phase and the persistence of increased replication origins activation in the next S-phase [79]. This result is consistent with our data in RKO daughter cells, in which we still observe RT advances accompanied by a reduction in DNA halo size and an increase in MCMs loading to chromatin at the G1/S transition (Figure 5). This indicates that the RKO next cell generation inherits a perturbed replication program at both the spatial and temporal level. Interestingly, in RPE-1 normal cells, we did not observe this phenomenon, suggesting that these cells are probably less sensitive to this same replication stress, as it was demonstrated by the lower S-phase checkpoint activation (Appendix A) and/or set up of an appropriate mechanism to avoid the inheritance of replication stress-linked problems. 

## 4. Materials and Methods

### 4.1. Cell Lines, Cell Culture, and Drugs

The six human cell lines were purchased from ATCC. Cells were grown in culture medium supplemented with 10% fetal bovine serum (Gibco Life Technologies A31608-02, Thermo Fisher Scientific, Waltham, MA, USA) at 37 °C, 5% CO_2_, and 5% O_2_. HCT116, U2OS, and RKO cell lines were grown in Dulbecco’s Modified Eagle’s Medium (DMEM, Gibco Life Technologies 31966021, Thermo Fisher Scientific, Waltham, MA, USA), MRC5-N cell line was grown in Minimum Essential Medium Eagle (MEM-aplha, Gibco Life Technologies 22561021, Thermo Fisher Scientific, Waltham, MA, USA), RPE-1 cell line was grown in Roswell Park Memorial Institute Media (RPMI, Gibco Life Technologies 61870044, Thermo Fisher Scientific, Waltham, MA, USA), and K562 in Iscove Modified Dulbecco Media (IMDM, Gibco Life Technologies 21980032, Thermo Fisher Scientific, Waltham, MA, USA) supplemented with decomplemented serum. Aphidicolin (Sigma-Aldrich AO781-1MG, Sigma-Aldrich, St. Louis, MO, USA) stock solution was diluted in DMSO (Sigma-Aldrich D8418-250mL, Sigma-Aldrich, St. Louis, MO, USA) and kept at −20 °C for a maximum of 2 months after first thawing. Cells were synchronized in G1/S with 0.5 mM L-Mimosine (Sigma-Aldrich M0253, Sigma-Aldrich, St. Louis, MO, USA) for 24 h.

### 4.2. Cell Lysis, Fractionation, and Western Blotting

For whole cell extract, cells were lysed for 30 min on ice with classic lysis buffer (0.3 M NaCl, 1% triton, 50 mM Tris pH7.5, 5 mM EDTA, 1 mM DTT, and 1X Halt protease and phosphatase inhibitor cocktail from Thermo Fisher Scientific 78445, Thermo Fisher Scientific, Waltham, MA, USA). For subcellular fractionation, cells were lysed in Buffer A (HEPES 10 mM pH 7.9, KCl 10 mM, MgCl_2_ 1.5 mM, sucrose 0.34 M, Glycerol 10%, dithiothreitol (DTT) 1 mM, 1X Halt protease and phosphatase inhibitor cocktail) complemented with Triton X-100 0.1% for 5 min on ice. After centrifugation at 1500 rcf, 5 min, 4 °C, the supernatant was clarified by high-speed centrifugation (18,000 rcf, 4 °C, 15 min) to obtain the cytoplasmic fraction. The pellet was washed once with Buffer A and then incubated in Buffer B (EDTA 3.2 mM, DTT 1 mM, Halt protease, and phosphatase inhibitor cocktail) for 30 min on ice. After centrifugation (1700 rcf, 5 min, 4 °C), the supernatant was collected as the soluble nuclear fraction. The pellet (chromatin fraction) was washed once with Buffer B and resuspended in the same buffer. Then, the whole cell extracts and chromatin-enriched fractions were sonicated (10 pulses of 1 s at 40% amplitude with a Sonics Vibra Cell Ultrasonic processor, Sonics & Materials, Inc., Newtown, CT, USA) and Laemmli buffer was added in order to have a final protein concentration of 2 and 0.5 µg/µL, respectively. The detection of pChk1 (S345, Cell signaling 2341, Rabbit, Cell Signaling Technology, Danvers, MA, USA), Chk1 (Santa Cruz sc 8408, Mouse, Santa Cruz Biotechnology, Inc., Santa Cruz, CA, USA), Actinin (MBL 05-384, Mouse, MBL International Corporation, Woburn, MA, USA), MCM2 (Abcam ab-4461, Rabbit, Abcam, Cambridge, MA, USA), p-MCM2 S40 (Abcam ab133243, Rabbit, Abcam, Cambridge, MA, USA), ORC2 (MBL M055-3, Mouse, MBL International Corporation, Woburn, MA, USA), histone H3 (Abcam ab1791, Rabbit, Abcam, Cambridge, MA, USA), Lamin A/C (Santa cruz sc-7293, Mouse, Santa Cruz Biotechnology, Inc., Santa Cruz, CA, USA), and α-Tubulin (Sigma T5168, Mouse, Sigma-Aldrich, St. Louis, MO, USA) was done by running SDS-page gels, transferring on PVDF membranes, blocking with 5% milk, incubating with primary antibody (in TBS-T, adapted dilutions, Bio-Rad, California, CA, USA) followed by secondary antibody (MBL 70765 Mouse or MBL 70745 Rabbit, MBL International Corporation, Woburn, MA, USA) and finally revealing thanks to ECL (Biorad 170-5161, Bio-Rad, California, CA, USA) under the ChemiDoc imaging system (Bio-Rad, California, CA, USA).

### 4.3. Replication Timing Analysis

Ten to 20 million of exponentially growing mammalian cells (with DMSO (Sigma-Aldrich D8418-250mL, Sigma-Aldrich, St. Louis, MO, USA) or aphidicolin (Sigma-Aldrich AO781-1MG, Sigma-Aldrich, St. Louis, MO, USA) were incubated with 0.5 mM BrdU (Abcam, #142567, Abcam, Cambridge, MA, USA), protected from light, at 37 °C for 90 min. After washing in PBS, cells were fixed in 75% final cold EtOH and stored at −20 °C. BrdU labeled cells were incubated with 80μg/mL Propidium Iodide (Invitrogen, P3566) and with 0.4 mg/mL RNaseA (Roche, 10109169001, Sigma-Aldrich, St. Louis, MO, USA) for 15 min at room temperature and 150,000 cells were sorted in early (S1) and late (S2) S-phase fractions using a Fluorescence Activated Cell Sorting system (FACSAria Fusion, Becton Dickinson, Franklin Lakes, NJ, USA) in Lysis Buffer (50 mM Tris pH = 8, 10 mM EDTA, 0.5% SDS, 300 mM NaCl) and stored at −20 °C until the following steps. DNA from S1 and S2 fractions of sorted cells was extracted using Proteinase K treatment (200 µg/mL, Thermo Scientific, EO0491, Thermo Fisher Scientific, Waltham, MA, USA) followed by phenol-chloroform extraction and sonicated to a size of 500–1000 base pairs (bps), as previously described [37]. Immunoprecipitation was performed using an IP star robot at 4 °C (indirect 200 µL method, SX-8G IP-Star^®^ Compact Automated System, Diagenode, Denville, NJ, USA) with an anti-BrdU antibody (10 μg, purified mouse Anti-BrdU, BD Biosciences, #347580, BD Biosciences, San Jose, CA, USA). Denatured DNA was incubated for 5 h with anti-BrdU antibodies in IP buffer (10 mM Tris pH = 8, 1 mM EDTA, 150 mM NaCl, 0.5% Triton X-100, 7 mM NaOH) followed by an incubation for 5 h with Dynabeads Protein G (Invitrogen, 10004D, Thermo Fisher Scientific, Waltham, MA, USA). Then, beads were washed with Wash Buffer (20 mM Tris pH = 8, 2 mM EDTA, 250 mM NaCl, 1% Triton X-100). Reversion was performed at 37 °C for 2 h with a solution containing 1% SDS, and 0.5 mg Proteinase K was followed, after beads removal, by an incubation at 65 °C for 6 h in the same solution. Immunoprecipitated BrdU-labeled DNA fragments were extracted with phenol-chloroform and precipitated with cold ethanol. Control quantitative PCRs (qPCRs) were performed using oligonucleotides specific of mitochondrial DNA, early (BMP1 gene), or late (DPPA2 gene) replicating regions [10,37]. Whole genome amplification was performed using SeqPlex^tm^ Enhanced DNA Amplification kit as described by the manufacturer (Sigma-Aldrich, SEQXE, Sigma-Aldrich, St. Louis, MO, USA). Amplified DNA was purified using PCR purification product kit as described by the manufacturer (Macherey-Nagel, 740609.50, MACHEREY-NAGEL, Dueren, Germany). DNA amount was measured using a Nanodrop. Quantitative PCRs using the oligonucleotides described above were performed to check whether the ratio between early and late replication regions was still maintained after amplification. Early and late nascent DNA fractions were labeled with Cy3-ULS and Cy5-ULS, respectively, using the ULS arrayCGH labeling Kit (Kreatech, EA-005, Leica Biosystems, Nussloch, Germany). Same amounts of early and late-labeled DNA were loaded on human DNA microarrays (SurePrint G3 Human CGH arrays, Agilent Technologies, G4449A, Agilent Technologies, Santa Clara, CA, USA). Hybridization was performed as previously described [37]. The following day, microarrays were scanned using an Agilent C-scanner with Feature Extraction 9.1 software (Agilent Technologies, Santa Clara, CA, USA). To determine the replication domains and do the comparative analysis in different conditions, the online platform specific for replication timing data START-R [38] was used, with biological duplicates for each condition. The differential analysis between DMSO and APH-treated cells is performed with the START-R suite with a defined *p*-value. The output bed file gave the list of significantly impacted genomic regions (ADVANCED or DELAYED) and the report of number and percentage of genomic regions impacted.

### 4.4. Hierachical Clustering Based on Replication Timing

We performed hierarchical clustering to aggregate cell lines with similar replication timing patterns. To do so, we followed the protocol published by Ryba et al. [84], using the “pvclus” package in R studio, with the code attached in Appendix A (R code for replication timing data clustering).

### 4.5. Genomic Studies of Advanced and Delayed Replication Timing Domains

For each experiment, START-R Analyzer [38] generates segmentation bed files corresponding to early, mid, late, advanced, and delayed replicating domains. All genomic studies were preformed using Galaxy website: https://galaxyproject.org, accessed on 6 May 2021 [85]. Using the chromosomal coordinates of these domains, we calculated their size and generated boxplots. To study the GC content of these domains, we extracted their genomic DNA sequences as a fasta format using the Extract Genomic DNA tools using coordinates from assembled/unassembled genomes and calculated the percentage of GC in each domain using the geecee EMBOSS tool (Version 5.0.0), which calculates fractional GC content of nucleic acid sequences. We collected the data and drew boxplots. The coverage of the different replicating domains with constitutive origins and genes was done with the Coverage tool (Version 1.0.0). Boxplots illustrating differences in these coverages were generated. The constitutive origins file (hg19 genome assembly) was taken from [86] and converted with LiftOver to hg18 genome assembly. The position of genes used for gene coverage came from the UCSC table browser RefSeq Genes database without duplicates and with hg18 genome assembly. 

### 4.6. aRTIL Intersections and Randomization Procedure 

To assess the overlap between aRTIL or between aRTIL and CFS, we used a randomization procedure. For each aRTIL list, we generated 1000 instances of randomized intervals on the mappable fraction of the human genome (hg19) using the bedtools shuffle function. Then, we calculated the Jaccard index between each aRTIL and target region (either other aRTIL or CFS regions) and did the same for the randomizations. We represented the Jaccard index for experimental data and randomization and report the z-score of the Jaccard index (defined by subtracting from each experimental value the mean of randomized values and then dividing by standard deviation of those values).

### 4.7. Gene Expression Microarrays

Exponentially growing cells (with DMSO or aphidicolin, Sigma-Aldrich, St. Louis, MO, USA) were harvested and RNAs were extracted with an RNeasy plus mini kit (QIAGEN Sciences Inc, Germantown, MD, USA). RNAs quality and quantity were controlled using Nanodrop ND-1000 and Bioanalyzer 2100 Expert from Agilent. cDNAs were prepared according to the standard ThermoFisher protocol from 100 ng total RNA (GeneChip™ WT PLUS Reagent Kit Manual Target Preparation for GeneChip™ Whole Transcript (WT) Expression Arrays User Guide, Thermo Fisher Scientific, Waltham, MA, USA). Following fragmentation, 5.5 µg of single-stranded cDNA were hybridized on Human Clariom S Arrays in GeneChip Hybridization Oven 645 for 16 hr at 45 °C. The arrays were washed and stained in the Affymetrix Fluidics Station 450. Arrays were scanned using the GeneChip Scanner GC3000 7G, and images were analyzed using Command Console software to obtain the raw data (values of fluorescent intensity). The data were analyzed with TAC (Transcriptome Analysis Console, version 4.0.2.15) from Thermo Fisher Scientific (Waltham, MA, USA). Microarrays were normalized with the “Robust Multichip Analysis” (SST-RMA) method. Statistical analysis allowed tagging of genes according to the fold change (FC) and the *p*-value adjusted together with ANOVA and Benjamini–Hochberg correction.

### 4.8. ATAC-seq

A total of 100,000 exponentially growing RKO cells were trypsinized, washed in PBS, and treated with 1:100 volume of RNase-free DNase (QIAGEN Sciences Inc., Germantown, MD, USA) and DMEM media for 30 min at 37 °C in the incubator. Cells were trypsynized, washed in PBS, and resuspended in 500 μL of ice-cold cryopreservation solution (50% FBS, 40% DMEM, 10% DMSO), transferred into a 2 mL cryotubes, and frozen in a pre-chilled Mr. Frosty container at −80 °C overnight or more before sending to Active Motif to perform ATAC-seq assay. Then, the cells were thawed in a 37 °C water bath, pelleted, washed with cold PBS, and tagmented as previously described [87], with some modifications based on [88]. Briefly, cell pellets were resuspended in lysis buffer, pelleted, and tagmented using the enzyme and buffer provided in the Nextera Library Prep Kit (Illumina, San Diego, CA, USA). Then, tagmented DNA was purified using the MinElute PCR purification kit (QIAGEN Sciences Inc, Germantown, MD, USA), amplified with 10 cycles of PCR, and purified using Agencourt AMPure SPRI beads (Beckman Coulter, Irving, TX, USA). Resulting material was quantified using the KAPA Library Quantification Kit for Illumina platforms (KAPA Biosystems, Sigma-Aldrich, St. Louis, MO, USA) and sequenced with PE42 sequencing on the NextSeq 500 sequencer (Illumina, San Diego, CA, USA). Analysis of ATAC-seq data was very similar to the analysis of ChIP-Seq data. Reads were aligned using the BWA algorithm (mem mode; default settings). Duplicate reads were removed, and only reads mapping as matched pairs and only uniquely mapped reads (mapping quality >= 1) were used for further analysis. Alignments were extended in silico at their 3′-ends to a length of 200 bp and assigned to 32-nt bins along the genome. The resulting histograms (genomic “signal maps”) were stored in bigWig files. Peaks were identified using the MACS 2.1.0 algorithm at a cutoff of *p*-value 1 × 10^−7^, without control file, and with the –nomodel option. Peaks that were on the ENCODE blacklist of known false ChIP-Seq peaks were removed. Signal maps and peak locations were used as input data to Active Motifs proprietary analysis program, which creates Excel tables containing detailed information on sample comparison, peak metrics, peak locations, and gene annotations. To annotate the ATAC-seq peak value and coverage within genomic regions of interest, we used Merge BedGraph and AnnotateBed bedtools functions, respectively [85] (Version 2.29.2). Then, we normalized the three biological replicates values across all genomic regions (Early, Mid, Late, ADV and DEL) by a 2-way ANOVA Sidak’s multiple comparisons test.

### 4.9. Processing ChIP-seq Data from Public Databases

ChIP-seq and pDamID, data were downloaded from the ENCODE, GEO and 4DN projects, respectively (Appendix A). The epigenetic marks coverage of each given regions list (Early, Mid, Late, ADV, DEL) was calculated using AnnotatedBed bedtools function [85] (Version 2.29.2). The mean for each epigenetic mark coverage in given regions was calculated to generate clustering tree heatmap based on Pearson correlations with ClustVis software [89].

### 4.10. Fluorescent DNA Halo

A total of 400,000 cells were harvested after synchronization and treated with nuclei buffer (10 mM Tris at pH 8, 3 mM MgCl2, 0.1 M NaCl, 0.3 M sucrose, protease inhibitors) plus 0.5% Nonidet P40 for 5–10 min on ice (depending on cell line). Nuclei were attached to coverslips using cytospin (1500–1800 rpm for 5–10 min, depending on cell line), stained with DAPI (2 mg/mL for 4 min), and immersed in a buffer containing 25 mM Tris (pH 8), 0.5 M NaCl, 0.2 mM MgCl2, 1 mM PMSF, and protease inhibitors for 1 min, then in Halo Buffer (10 mM Tris at pH 8, 2 M NaCl, 10 mM ethylene diamine tetra acetic acid (EDTA), 1 mM DTT, protease inhibitors) for 4 min. After two washing steps with wash buffer 1 (25 mM Tris (pH 8), 0.2 M NaCl, and 0.2 mM MgCl2) for 1 min, and with buffer 2 (buffer 1 without NaCl) for 1 min extracted nuclei were fixed in 2% formaldehyde for 10 min and processed for immunofluorescence. Images containing about 200 halos per condition acquired with a Nikon Ni-E microscope and a DS-Qi2 camera with 64× objective and MFHR (Maximum Fluorescence halo Radius) was measured in Image J software.

## Figures and Tables

**Figure 1 ijms-22-04959-f001:**
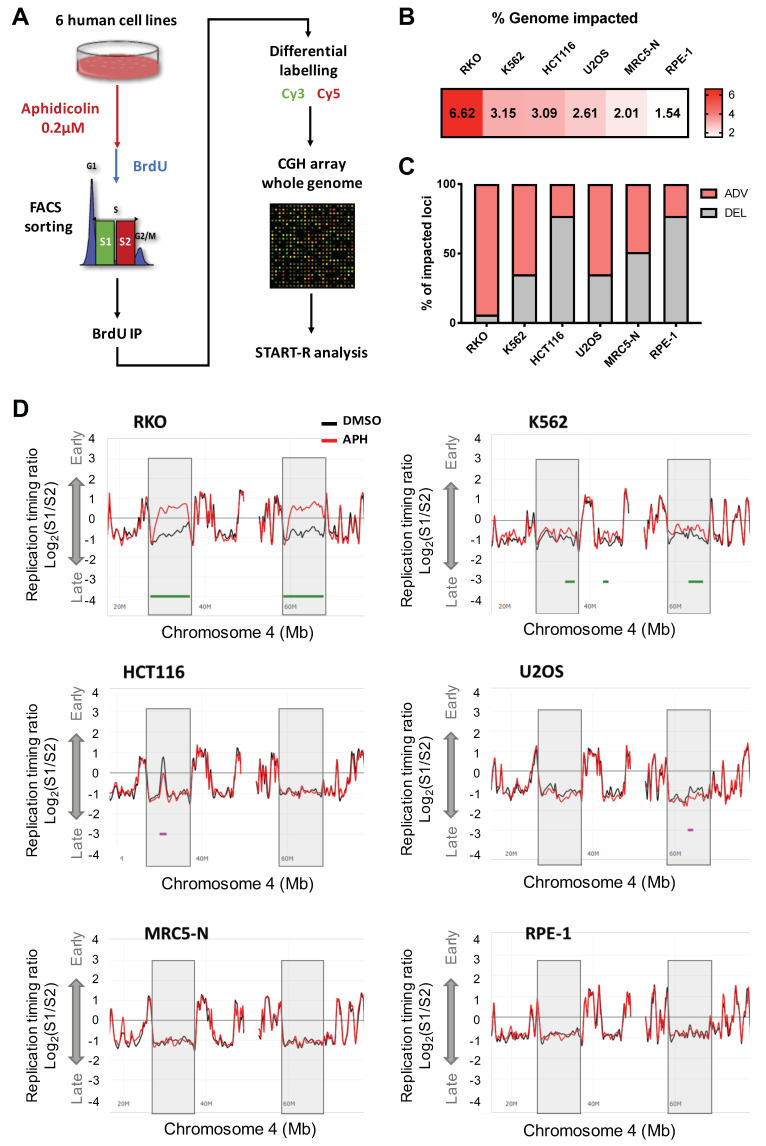
Low replication stress induces cell-type-specific replication timing changes. (**A**) Experimental protocol to assess whole genome-RT. Briefly, cells were treated with 0.2µM APH or DMSO during a time adapted for each cell line (see Appendix A), cells were labeled by BrdU during 1 h 30 and FACS sorted into two S-phase fractions (S1 for early S and S2 for late S). Then, DNA from each fraction is immunoprecipitated using a BrdU antibody and immunolabeled with Cy3 (S1) and Cy5 (S2) and then hybridized in whole human genome arrays. (**B**) Heatmap representing the coverage of the RT impacted genome (%) by APH treatment for each cell line. (**C**) Stacked histogram representing the proportion (%) of impacted loci or aRTIL (ADV in light pink and DEL in light gray). (**D**) START-R snapshots of Loess-smooth RT profiles in the same region (Chromosome 4) for the six cell lines. The dark lines correspond to replication timing of control (DMSO) and the red lines are replication timing of APH treated cells. Two regions of interest are highlighted in gray, significant ADV aRTIL are underlined in light green, and DEL aRTIL are in light pink.

**Figure 2 ijms-22-04959-f002:**
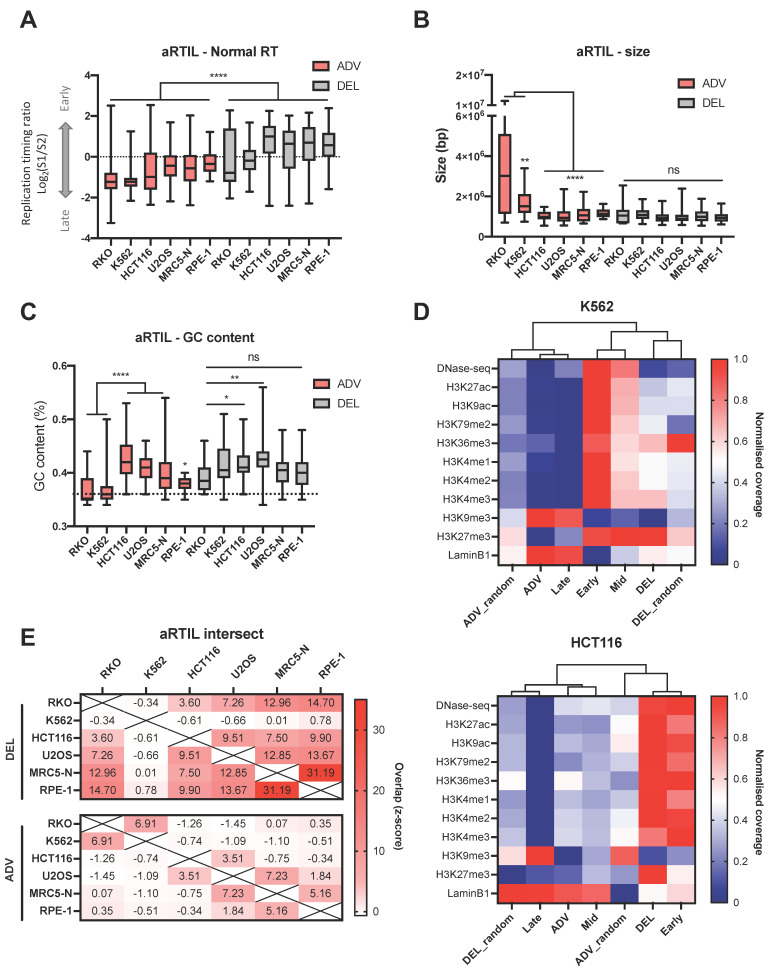
Genomic and epigenomic characteristics of aRTIL. Box and whiskers (min to max) representing (**A**) the normal replication timing (DMSO condition) (**B**) the size and (**C**) the GC content of ADV and DEL aRTIL in the six cell lines. Statistics: Kruskal–Wallis test with Dunn’s multiple comparison: **** *p* < 0.0001, ** *p* < 0.01, * *p* < 0.05, ns *p* > 0.05. (**D**) Heatmap of coverage and clustering trees based on Pearson correlations for epigenetic marks in K562 (top) and HCT116 (bottom) cell lines in different loci (Early, Mid, Late, ADV, DEL, ADV random, and DEL random). The coverage was normalized to have all values between 0 (blue) and 1 (red). (**E**) Heatmap of intersections between ADV and DEL aRTIL in between the six cell lines. Jaccard index was used to measure the intersection, and the z-score between Jaccard for aRTIL list and randomized regions is reported (see method section).

**Figure 3 ijms-22-04959-f003:**
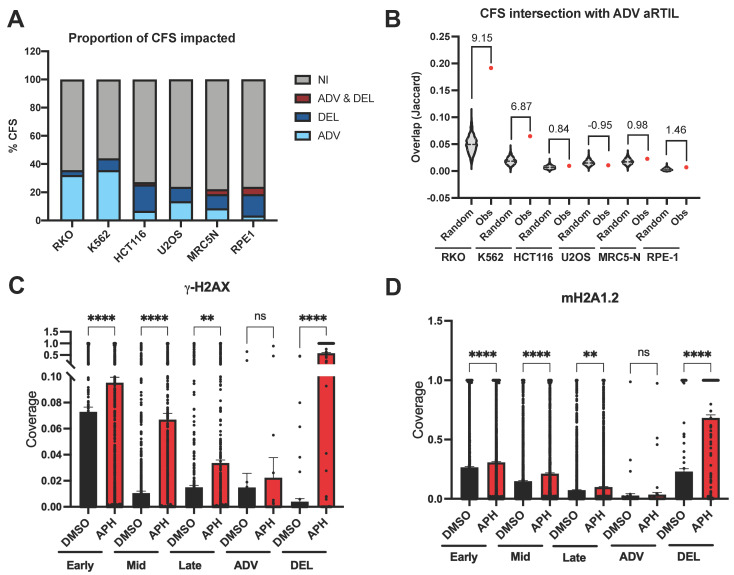
ADV aRTIL are closely related to CFS but are not enriched in DNA damage signaling histone marks. (**A**) Stacked histogram representing the proportion (in %) of CFS impacted by aphidicolin (ADV or DEL). (**B**) Jaccard index for overlap of CFS with observed ADV aRTIL (red) and randomly shuffled regions of equal size (gray). Values of z-score between random and observed Jaccard are indicated as statistics. Scatter plots with bar (SEM) representing the coverage of (**C**) γ-H2AX histone mark and (**D**) mH2A1.2 histone mark (ChIP-seq data for K562 cell line) on early, mid, late, ADV aRTIL, and DEL aRTIL genomic regions in DMSO (black) and APH (red). Statistics: Kruskal–Wallis with Dunn’s multiple comparison: **** *p* < 0.0001, ** *p* < 0.01, ns *p* > 0.05.

**Figure 4 ijms-22-04959-f004:**
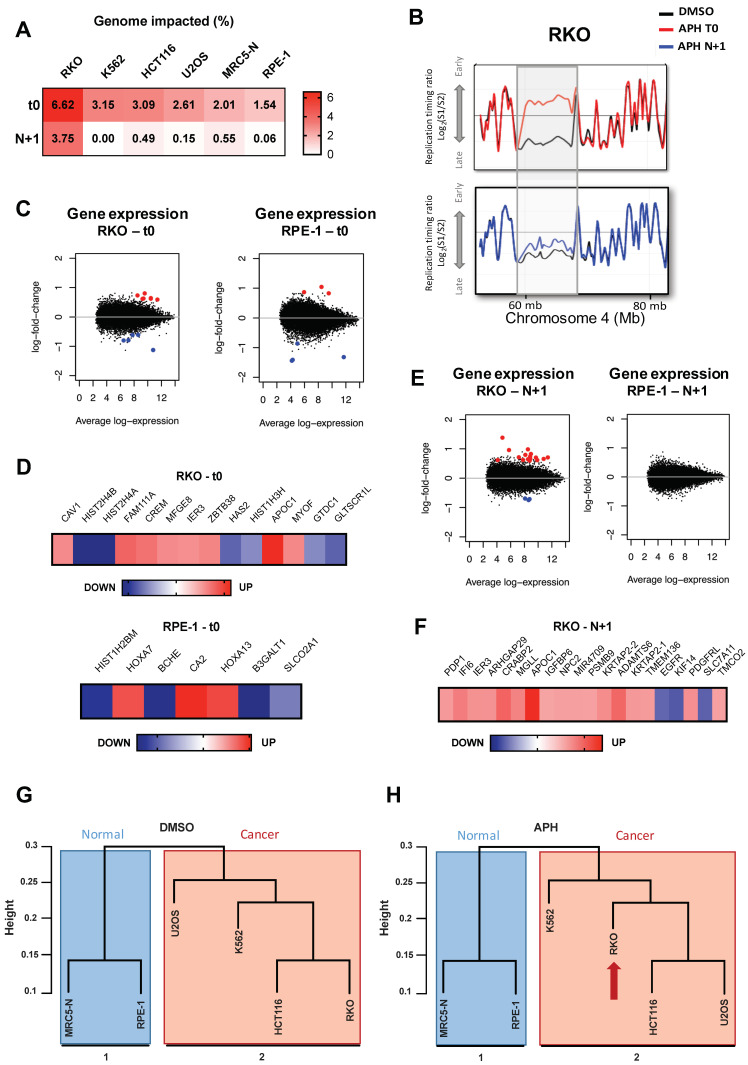
ADV aRTIL can be transmitted to daughter cells. (**A**) Heatmap representing the coverage (in %) of RT impacted genome in mother cells (t0) and released daughter cells (N + 1) for the six cell lines. (**B**) START-R snapshots of Loess-smooth replication timing profiles for the same region (Chromosome 4), in RKO mother (APH t0) and daughter cells (APH N + 1). The dark lines correspond to replication timing of control (DMSO), the red line is replication timing of t0 APH-treated cells, and the blue line is the replication timing of N + 1 daughter cells released from APH treatment. (**C**) Scatterplots for differential gene expression between DMSO and APH treatment from microarray data in RKO (left) and RPE-1 (right) mother cells (t0). Significantly APH UP genes are marked in red and APH DOWN genes are marked in blue. Statistics: ANOVA, Benjamini–Hochberg correction, *FDR* < 0.05. (**D**) Heatmaps with names and color scale based on log2 fold change for the significantly impacted genes in RKO (top) and RPE-1 (bottom) mother cells (t0), APH DOWN, in blue; APH UP, in red. (**E**) Scatterplots for differential gene expression between DMSO and APH treatment from microarray data in RKO (left) and RPE-1 (right) daughter cells (N + 1). Significantly APH UP genes are marked in red and APH DOWN genes are marked in blue. Statistics: ANOVA, Benjamini–Hochberg correction, *FDR* < 0.05. (**F**) Heatmap with names and color scale based on log2 fold change for the significantly impacted genes in RKO daughter cells (N + 1), APH DOWN, in blue; APH UP, in red. Statistics: ANOVA, Benjamini–Hochberg correction, *FDR* < 0.05. Cluster dendrograms aggregating cell lines based on replication timing patterns in (**G**) DMSO and (**H**) APH conditions. Distance: correlation and clustering method: average (see method).

**Figure 5 ijms-22-04959-f005:**
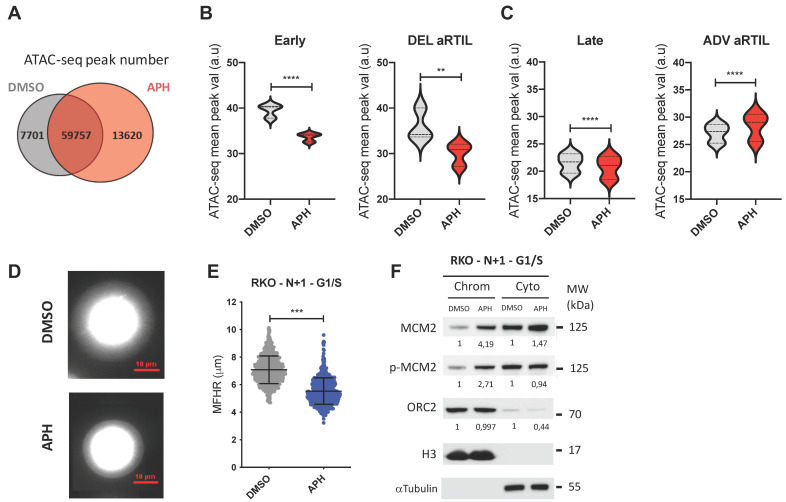
Aphidicolin modulates chromatin accessibility in RKO cells. (**A**) Venn diagram of ATAC-seq peak number for DMSO and APH-treated conditions. (**B**) Violin plots representing the comparison of ATAC-seq peak value within early control regions (left) and DEL aRTIL (right) between DMSO (gray) and APH (red) conditions. Statistics: Wilcoxon matched-pairs signed rank test **** *p* < 0.0001, ** *p* < 0.01. (**C**) Violin plots representing the comparison of ATAC-seq peak value within late control regions (left) and ADV aRTIL (right) between DMSO (gray) and APH (red) conditions. Statistics: Wilcoxon matched-pairs signed rank test **** *p* < 0.0001. (**D**) Visualization (red scale = 10 µm) and (**E**) Quantification of DNA Halo size (MFHR in µm) in RKO G1/S synchronized daughter cells released from DMSO or aphidicolin treatment. Error bars: SD, Statistics (N = 3): Unpaired *t*-test, Welch’s correction *** *p* < 0.001. (**F**) Western blot on chromatin (Chrom) and cytoplasmic (Cyto) protein fractions to quantify the amount of MCM2, p-MCM2, and ORC2 in cells synchronized in G1/S with L-mimosine. The mean fold change between DMSO and APH condition measured in three independent experiments is reported on the figure. H3 and tubulin values were used to normalize protein signal in chromatin and cytoplasmic fraction, respectively.

**Figure 6 ijms-22-04959-f006:**
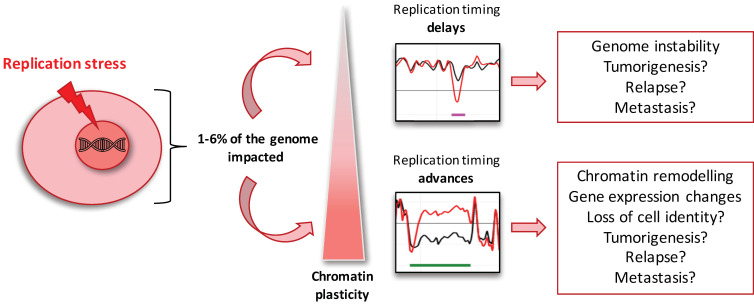
Model for the differential impact of low replication stress on replication timing. Scheme of the proposed response of cells to low replication stress depending on the level of chromatin plasticity. Replication stress would differentially impact the cells depending on their background. Cells with chromatin plasticity would have a higher proportion of RT advances under replication allowing DNA replication program modifications and the regulation of specific genes’ expression, potentially affecting cellular identity. Importantly, this work paves the way for future studies investigating molecular effectors involved in chromatin accessibility and replication timing changes in response to stress. These new targets could have a great potential to prevent cancer cells’ adaptation to replication stress, often leading to therapy resistance.

## Data Availability

DNA replication timing, ATAC-seq and microarray gene expression data are available under accession numbers GSE156618, GSE156552 and GSE156521, respectively.

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
