# Peer review of "Low Replicative Stress Triggers Cell-Type Specific Inheritable Advanced Replication Timing"

_ijms, 2021, doi:10.3390/ijms22094959_

Round 1

Reviewer 1 Report

In this article Courtot et al measured RT  in duplicates in 6 cell lines treated or not with  mild aphidicolin treatment. They defined differential regions, three of which they experimentally validated in RKO cells. They then characterized these regions through a variety of genomic parameters, concluding that cells undergo lineage-specific changes in RT program as a result of aphidicolin treatment. These changes are found to be partially maintained in the next generation, mostly in RKO cells. They also found overlap between these differential regions and the regions whose chromatin accessibility changes under replication stress. Overall, the paper tackles an important question regarding the changes in the RT program upon mild replication stress, yet I cannot recommend publication in IJMS for three reasons. First, I was not convinced that all the reported RT changes are correct and many of them may be measurement errors. Secondly, the meaning of the results is not clear, especially since they are not consistent with previous knowledge and expectation (for example the lack of overlap between CFSs and delayed RT regions). Finally, there are many places in which the paper is not written well,  both in terms of properly supporting the claims with the data and in terms of statistical analyses. The overall impression that I got from the paper is that mild aphidicolin treatment has a significant effect on RKO cells (in terms of advanced RT, decreased Halos size and MCM binding), but has a minor to no effect on other cell lines. This conclusion is quite different than the authors’ conclusions.

Specific comments:

A prerequisite for any genomic paper is to first convince the reader of the validity of the results. This should be done by showing consistency between repeats, consistency with the literature and by validating some of the results with a different method. These were done to some extent for RKO findings (which are much more dominant than all the other findings both in terms of the size of the regions and the amplitude of the change) but was not done for the more minor changes in the other cell lines. I would suggest, for example, showing for each differential region a comparison between the two repeats (this can be done by showing a histogram of the difference between them in the change in the RT). The authors partially addressed this concern by comparing the results with known genomic features in figure S4, however, these results are quite trivial since RT there is a higher chance that random changes in an early region will be delays, and vice versa. Thus, the results in figures S4 and 2a, c, d are consequences of this and do not necessarily confirm the validity of the differential regions. My concern regarding the small RT changes observed in most of the cell lines is partially due to the algorithm they used for differential RT calling. They used their START-R algorithm which uses the Holm-hypothesis correction for multiple hypothesis correction, however in order to evaluate it more information is needed (for example the number of hypothesis considered in each case).

Figure S1c demonstrates that the length of the aphidicolin treatment was accurate since after its completion very few cells were in G1. Given that the aphidicolin treatment was quite long (table S2) it is hard to understand the results. With 16 h of BrdU staining (for RKO) the only possibility for a lack of BrdU positive cells in G1 is a very long G2 phase (almost 16 hours), which contradicts the results shown in  figure S1-D and E.

Figure S4 - the Early, mid and late comparison is trivial and not relevant to this paper. The comparisons are done for some cell lines with the advanced and for others with the delayed without a good rational. Moreover, the claim in the text is about the similarity of the delayed regions to early/mid regions and this cannot be supported by a comparison with the advanced regions as was done for RKO and K562. S4D is not referred to in the text. It shows that the size of domain of these variable regions is generally small, which supports the concern that they are just noise. The K562 ADV and DEL regions have different sizes to each other (like those of RKO), making this cell line's changes more credible.

Line 161: The claim that there exists 'greater variability for cancer cells' is not clear and appears not to be supported by S4A-C

Figure 2B- it is very hard to evaluate this figure because of the lack of the color key. Nevertheless there are some obvious problems with it, especially cases in which the random differentially regions show higher overlap than the real differential regions (i.e. K562 delayed with U2OS and HCT116 and RKO advanced  with almost all other cell lines).

Figure 2C – here again the randomization seems to be problematic since random regions should behave similarly regardless of their advanced or delayed annotation. Yet it seems that the random regions are clustered together with their origins at least in k562. In addition, it is not clear why this analysis was performed only on two cell lines and the term normalized coverage should be defined.

Figure 2D – Adds nothing to the paper.

Figure S5-E We assume that the normal RT map is being shown. This is meaningless without also displaying the perturbed RT on the same figure.

Figure 3B- A statistical test should be added. Also, it is maitakedly called a historgram in the legend.

Figure 4 B – nicely shows that there are differential regions only in the RKO cell line in the next cell cycle, however we cannot tell whether the differential regions in t0 and in N+1 overlap. In 4C there is one nice example of an overlap, but without a proper demonstration that most of them overlap one cannot conclude that the RT changes are maintained in the next generation.

Figure 4 D-G- the same concern is valid here - do the genes affected by aphidicolin in t0 and N+1 overlap? The increase from 14 genes to 20 suggests the possibility that these are different genes. This is easy to check, but is vital for the article. If they are demonstrably different genes, then the whole point is negated.

Figure 4 H+I. How did they create the ’cluster dendogram’? In the legend it is written that the clustering is based on “p-values reflecting on RT signatures in DMSO”, and I cannot figure out what this means. In addition, the meaning of the changes in the cluster structure are not clear.

Figure 5- The massive changes in ATAC-seq with APH is peculiar considering the very mild change in RT and expression. In Figures 5 A, E and G we are surprised by the extent of the effect shown, given the 6% total effect on RT. Suggesting that both the ATAC results and the Halo and MCM results are associated with each other and are not associated to the mild changes in RT.

Statistical analyses - The frequent use of two way ANOVA tests (e.g. Figure 3C,D,Figure 4 D,E,F,G) is incorrect. Multiple hypothesis adjustment (e.g. 4 D,E,F,G) should be mentioned, when done. What method was used?

Minor points:

Lacking colors in key of figure, e.g. 2B, 2C, 2D

Figure S4 – what is the difference between NA and NS?

Lines 201:202 – ' Non-random overlap of DEL aRTIL and ADV aRTIL with CFS was investigated (Overlap Jaccard, Figure 2B).' I think that the reference should be to 3B.

3E adds nothing after we have seen the 3c+D.

S9-B what are C1-5?

Figure 5 B+C are redundant with 5A.

The final sentence of the results is exaggerated and should probably be toned down a bit in order to reflect the results.

Line 356: please define 'chromatin flexibility of RKO cells'.

Author Response

Reviewer 1

Comments and Suggestions for Authors

Dear reviewer 1,

We noticed that reviewer 1 does not recommend our manuscript for publication in IJMS.  The Editor offered us the chance to respond to his comments within 7 days and to improve the manuscript in consequence for a second review. We really hope that our answers will change the opinion of reviewer 1. Reviewer 1 overall impression from the paper is “that mild aphidicolin treatment has a significant effect on RKO cells (in terms of advanced RT, decreased Halos size and MCM binding), but has a minor to no effect on other cell lines”. We agree with this comment, that’s why we focus the main part of our work on this cell line. Nevertheless, we analyzed RT in response to low replication stress on 6 cell lines and we thought that the presentation of the results obtained on all these lines highlight the specificity of RKO cells.

Therefore, we modified the part describing aRTIL genomic and epigenomic features by deleting Figure S4 and focusing on the specific characteristic of ADV aRTIL found in RKO and K562.

Reviewer 1 questions the method of analysis of timing changes with the START-R algorithm, and “suspects the validity of the results obtained, except for ADV aRTIL found in RKO and K562 that we validated with a second method”.

We used STAR-R with all the recommendations that have been validated for the same of this type of analysis in the article (Hadjadj, D. NAR Genom Bioinform 2020) and we applied the Mean method with a p-value < 0,05 to significantly detect the disturbs in the replication timing between two conditions on experimental replicates.  This method allows us to be confident about the results we obtained.

Finally, we really apologize for all the problems of conversion of the files on the quality of the figures and the absence of colors on certain parts.

In this article Courtot et al measured RT in duplicates in 6 cell lines treated or not with mild aphidicolin treatment. They defined differential regions, three of which they experimentally validated in RKO cells. They then characterized these regions through a variety of genomic parameters, concluding that cells undergo lineage-specific changes in RT program as a result of aphidicolin treatment. These changes are found to be partially maintained in the next generation, mostly in RKO cells. They also found overlap between these differential regions and the regions whose chromatin accessibility changes under replication stress. Overall, the paper tackles an important question regarding the changes in the RT program upon mild replication stress, yet I cannot recommend publication in IJMS for three reasons. First, I was not convinced that all the reported RT changes are correct and many of them may be measurement errors. Secondly, the meaning of the results is not clear, especially since they are not consistent with previous knowledge and expectation (for example the lack of overlap between CFSs and delayed RT regions).

As shown in figure 3A, a portion of CFS contains DEL domains for all the cell lines and mainly for HCT116, MRC5-N and RPE-1. However, the Jaccard index, a statistic used in understanding the similarities between sample sets, did not show a significant overlap between CFS and DELs compared to randomized regions. This indicate that DELs can lie in some CFS but are not specifically enriched in these regions. One explanation for the lack of significant overlap between CFS and Delayed regions in our study is that the late S phase fraction (S2) does not cover the G2 phase so we didn’t get cells with under-replicated CFS after aphidicolin treatment which have been described to have a delayed RT in response to aphidicolin. In the text L 190-203

As suggested by reviewer 2, we added large gene coverage (>400kb) quantification in aRTIL that we previously have done (fig S5B), and description in the manuscript L 190-203. This additional analysis show that large genes are exclusively associated to DEL aRTIL in HCT116, MRC5-N and RPE-1 cells, that also have high level of DEL aRTIL and DEL aRTIL identified within CFS genomic regions. Large genes also have been identified in ADV aRTIL from RKO, K562 and U20S cells.

Even if our results do not allow us to firmly conclude about DEL aRTIL, they confirm that CFS can harbour RT advances in RKO and K562 cell lines.

Finally, there are many places in which the paper is not written well, both in terms of properly supporting the claims with the data and in terms of statistical analyses.

We tried to improve the manuscript and reviewed our conclusions. More explanation and details about statistical analyses have been added at the specific figure’s legends.  Some of statistical analyses have been changed following reviewer 1 recommendations (Fig 3)

Specific comments:

A prerequisite for any genomic paper is to first convince the reader of the validity of the results. This should be done by showing consistency between repeats, consistency with the literature and by validating some of the results with a different method. These were done to some extent for RKO findings (which are much more dominant than all the other findings both in terms of the size of the regions and the amplitude of the change) but was not done for the more minor changes in the other cell lines. I would suggest, for example, showing for each differential region a comparison between the two repeats (this can be done by showing a histogram of the difference between them in the change in the RT).

We validated ADV in RKO because it was totally new and unexpected to have advanced RT in response to aphidicolin, a drug that inhibits replicative ADN polymerases and slows down replication forks.  On the contrary, DEL were more expected and several publications already described RT delays in response to low doses of aphidicolin (Letessier A. Nature 2011; Brison O. et al. Nat Com 2019; Sarni D. et al. Nat Com 2020). As we decided to focus our attention and further analyses specifically on ADV aRTIL found in RKO cells, we found not essential to validate DEL aRTIL by a second method. Nevertheless, RT was analyzed using START-R algorithm that was validated on several sets of RT experiments already published (Hadjadj, D. NAR Genom Bioinform 2020). Finally, START-R analyses on replicates that give us the significant RT modifications were done with the Holm procedure that is currently used for microarrays (Hadjadj, D. NAR Genom Bioinform 2020). Importantly, we applied the Mean method with a p-value < 0,05 to significantly detect the disturbs in the replication timing between two conditions in experimental replicates. We also have shown in figure 2E and S4D that DEL aRTIL are often shared between cells lines indicating that they are not randomly distributed as could be the case for measurement errors.

Finally, to convince the reviewer 1 of the reproducibility of our experiments, the figure below shows the replicates for 3 HCT116 and MRC5-N DEL aRTIL, all in Chromosome 3. DMSO is in blue and APH in red. A pink track identified DEL aRTIL that START-R determined in the replicate’s analyses.

The authors partially addressed this concern by comparing the results with known genomic features in figure S4, however, these results are quite trivial since RT there is a higher chance that random changes in an early region will be delays, and vice versa. Thus, the results in figures S4 and 2a, c, d are consequences of this and do not necessarily confirm the validity of the differential regions.

 As you suggest below that the results in Figure S4 are not relevant to this paper we remove it and we modified the corresponding paragraph, L 153-172.

 My concern regarding the small RT changes observed in most of the cell lines is partially due to the algorithm they used for differential RT calling. They used their START-R algorithm which uses the Holm-hypothesis correction for multiple hypothesis correction, however in order to evaluate it more information is needed (for example the number of hypothesis considered in each case).

The Bonferroni and Holm procedure are both proposed by START-R as well as the procedure of Hochberg, Hommel and FDR.

The Holm, Hochberg, Hommel and Benjamini & Hochberg methods are designed to give strong control of the family-wise error rate. There seems no reason to use the unmodified Bonferroni correction because it is dominated by Holm's method, which is also valid under arbitrary assumptions.

- Hochberg's and Hommel's methods are valid when the hypothesis tests are independent or when they are non-negatively associated (Sarkar, 1998; Sarkar and Chang, 1997). Hommel's method is more powerful than Hochberg's, but the difference is usually small and the Hochberg p-values are faster to compute.

We choose the Holm procedure because the method was validated in the referent publication on several sets of RT experiments already published (Hadjadj, D. NAR Genom Bioinform 2020).

Figure S1c demonstrates that the length of the aphidicolin treatment was accurate since after its completion very few cells were in G1. Given that the aphidicolin treatment was quite long (table S2) it is hard to understand the results. With 16 h of BrdU staining (for RKO) the only possibility for a lack of BrdU positive cells in G1 is a very long G2 phase (almost 16 hours), which contradicts the results shown in figure S1-D and E.

Indeed, we observed a modest accumulation in G2/M but we saw a strong accumulation in S phase (figure S1D-E) indicating that aphidicolin treatment induce a longer S phase in RKO cells and would explain that treating cells for 16h is needed to see some of them reaching the next G1 phase.

Figure S4 - the Early, mid and late comparison is trivial and not relevant to this paper. The comparisons are done for some cell lines with the advanced and for others with the delayed without a good rational.

This figure has been removed and the corresponding text modified, L153-172.

Moreover, the claim in the text is about the similarity of the delayed regions to early/mid regions and this cannot be supported by a comparison with the advanced regions as was done for RKO and K562.

S4D is not referred to in the text. It shows that the size of domain of these variable regions is generally small, which supports the concern that they are just noise. The K562 ADV and DEL regions have different sizes to each other (like those of RKO), making this cell line's changes more credible.

Figure S4D has been moved to Figure 2B: The sizes appear small on the graph but correspond on average to 1M base pairs. Knowing that there is an average of one probe every 13Kb which makes 77 probes in the region. This number is more than sufficient to measure differences via the microarray approach. In CGH and transcriptome chips, the Bonferroni or Holm procedures allow the detection of differences with fewer probes, which have since been widely validated. Moreover, similar size of differential RT regions has been reported in the literature for DELs but also ADV (for examples: Sarni D et al. Nat. Com 2020; Rivera-Mulia J.C. et al. PNAS 2017)

Line 161: The claim that there exists 'greater variability for cancer cells' is not clear and appears not to be supported by S4A-C

As this figure has been removed, this sentence has been depleted.

Figure 2B- it is very hard to evaluate this figure because of the lack of the color key. Nevertheless, there are some obvious problems with it, especially cases in which the random differentially regions show higher overlap than the real differential regions (i.e. K562 delayed with U2OS and HCT116 and RKO advanced with almost all other cell lines).

We are again sincerely sorry for the lack of the color Key, and we understand that this may have made the interpretation of the results problematic

Figure 2B is now Figure 2E: your interpretation is exact and conform to what we say in the text L166-169 (This analysis shows that DEL aRTIL are non-randomly shared between all the cell lines, excepted the K562, in which DEL aRTIL seem to be more cell type specific.) and L180-181-182-193 (For ADV aRTIL we identified a significant overlap between RKO and K562 ADV aRTIL compared to random (Figure 2E bottom and Figure S4D). To a lesser extent, we also noticed significant overlap between MRC5-N and U2OS ADV aRTIL while in HCT116 and RPE-1 cell lines these latest are more cell type specific (Figure 2E bottom)).

 When random is higher than observed this mean that overlap is not significant. For example, in K562 cell line, RT delayed domains are not significantly found in other cell lines. On the contrary, we saw that ADV aRTIL from RKO are significantly found at the same place in K562 cells because the value of the Jaccard index is higher in the observed experiment than in the random analyses. These results led us to conclude that RT modifications in response to aphidicolin are cell type specific and that a specific signature of ADV aRTIL are shared between RKO and K562.

In addition, it is not clear why this analysis was performed only on two cell lines

All available epigenetics data were only for these two cell lines.

and the term normalized coverage should be defined.

Normalized coverage: max cov =1, min cov = 0

Figure 2D – Adds nothing to the paper.

We moved this figure to supp data (Figure S4C)

Figure S5-E We assume that the normal RT map is being shown. This is meaningless without also displaying the perturbed RT on the same figure.

Figure S5E is now Figure S4D. As you might have guessed the normal RT map is being shown (more information has been now added to the figure legend). By this illustration, we wanted to illustrate the normal timing of the impacted domains and also show that there are more intersections between the ADV RKO and K562 and more between DELs of the others cell lines in complement to Figure 2E

Figure 3B- A statistical test should be added. Also, it is maitakedly called a historgram in the legend.

 The Jaccard index, also known as the Jaccard similarity coefficient, is a statistic used for gauging the similarity and diversity of sample sets. This approach was applied for a similar purpose by Kim et al. (Mol Cell, 2018). “Histogram” in the legend has been removed.

Figure 4 B – nicely shows that there are differential regions only in the RKO cell line in the next cell cycle, however we cannot tell whether the differential regions in t0 and in N+1 overlap. In 4C there is one nice example of an overlap, but without a proper demonstration that most of them overlap one cannot conclude that the RT changes are maintained in the next generation.

This information was indicated in the text (L 244, 247): “Strikingly, in RKO cells, while DEL aRTIL returned to normal RT, the majority (28 of 49, 57%) of the strongest and largest ADV aRTIL were transmitted to the next cell generation, albeit with lower amplitude”.

If needed we can make a figure for the representation of this result. 

Figure 4 D-G- the same concern is valid here - do the genes affected by aphidicolin in t0 and N+1 overlap? The increase from 14 genes to 20 suggests the possibility that these are different genes. This is easy to check, but is vital for the article. If they are demonstrably different genes, then the whole point is negated.

Figure S8B and D show the list of genes whose expression was changed by aphidicolin at T0 and in N+1 daughter cells at the transcriptional level. Impacted genes are different at T0 and N+1. At first sight, this could minimize its interest, but we have seen that the genes impacted at T0 are rather associated with a stress response (Gene Ontology analyses in figS7B), whereas the genes impacted at N+1 are known to be deregulated in cancers, particularly of the colon cancer (table S3).This indicate that RT and gene expression changes are concomitant events in response to APH at T0 and that ADV aRTIL, related to increased chromatin accessibility can interfere with the correct setting up of replication and transcriptional program of the N+1 daughter cells. This concern has been discussed L 401-418.

Figure 4 H+I. How did they create the ’cluster dendogram’? In the legend it is written that the clustering is based on “p-values reflecting on RT signatures in DMSO”, and I cannot figure out what this means. In addition, the meaning of the changes in the cluster structure are not clear.

We have added a description of the method in the material and method part (L528-532) and added the R script in supplementary files.

As RKO and HCT116 are two colon cancer cell lines, it was expected to see them clustering in the DMSO condition. The fact that they are further apart in the APH condition reveals that the impact of aphidicolin in RKO affect the identity of the cells, as they do not anymore aggregate with the other colon cancer line HCT116, in which RT modifications are much lower.

Figure 5- The massive changes in ATAC-seq with APH is peculiar considering the very mild change in RT and expression. In Figures 5 A, E and G we are surprised by the extent of the effect shown, given the 6% total effect on RT. Suggesting that both the ATAC results and the Halo and MCM results are associated with each other and are not associated to the mild changes in RT.

ATAC seq changes in response to aphidicolin seems to be massive in term of global Peak number but as reported in Figure 5A, the mean peak value is significantly lower in all parts of the genome (early, late and DEL aRTIL domains) but higher in ADV aRTIL, showing that some chromatin accessibility modifications are specific to ADV aRTIL.

Statistical analyses - The frequent use of two way ANOVA tests (e.g. Figure 3C,D,Figure 4 D,E,F,G) is incorrect. Multiple hypothesis adjustment (e.g. 4 D,E,F,G) should be mentioned, when done. What method was used?

More information about statistical analyses have been added to the figure legends.

Minor points:

Lacking colors in key of figure, e.g. 2B, 2C, 2D

We apologize for this inconvenience, this has been corrected

Figure S4 – what is the difference between NA and NS?

This figure has been removed.

Lines 201:202 – ' Non-random overlap of DEL aRTIL and ADV aRTIL with CFS was investigated (Overlap Jaccard, Figure 2B).' I think that the reference should be to 3B.

Thank you for this correction, it has been changed

3E adds nothing after we have seen the 3c+D.

This panel has been removed

S9-B what are C1-5?

C1 to C5 are clusters that have been generated for quality control. They have no specific indications for the figure message.

Figure 5 B+C are redundant with 5A.

Figure 5 B+C give the statistical analyses so we propose to remove the 5A (heatmap) instead.

The final sentence of the results is exaggerated and should probably be toned down a bit in order to reflect the results.

This sentence has been changed.

Line 356: please define 'chromatin flexibility of RKO cells'.

Chromatin flexibility/plasticity is well studied during development and is involved in cellular identity. The concept of chromatin flexibility or plasticity in RKO cells indicate that chromatin is more dynamic in this cell type, mainly in response to replication stress. It is suggested by higher chromatin accessibility but also RT modifications that can be transmitted and gene expression changes L 441-442.

Reviewer 2 Report

In the manuscript “Low replicative stress triggers cell-type specific inheritable advanced replication timing” Courtot et al present data describing changes in replication timing after replication stress. The work is based on genome wide analysis of replication timing and the correlation to other genomic features. The work shows that replication timing is differentially affected after replication stress in the six different cell lines analysed. Moreover, the work finds that certain large regions in two of the cell lines show advanced replication timing after replication stress.

This work is purely descriptive and suffers from a lack of mechanistic insight. Nevertheless, it addresses a highly interesting subject, most of the experiments are well performed, and the data is clearly presented in the manuscript.

Major points.

  1. In Sup fig 1 there is no effect on CHK1-P of the APH treatment used in MRC5-N cells and RPE-1 cells (and minimal effect in U2OS and HCT116). Thus, the only cells that seems to have a clear response are also those with highest percent of genome affected. Therefore, the difference between the cell lines in this regard (Fig. 1B and may Fig 1C) is probably due to different levels of replication stress.

The authors cannot state in line 98 that replication stress is validated by Chk1 phosphorylation and they should in general be careful not to push their argumentation about cell line differences in inherited RT effects too much.

  1. In Fig. 3 how are CFSs defined? It is not clear. It is well established that CFSs are cell type specific, therefore this must be taken into consideration for each of the six cell lines. The authors should correlate ADV and DEL to giant genes (for instance larger than 1 megabase) that are known to be expressed in the specific cell line. This would provide a much more specific analysis of CFSs replication timing after replication stress. This could give important insight given the well-established effects of APH induced replication stress on CFSs.

Minor points.

Introduction line 46 and 48: The A and B compartment should be introduced better.

Introduction line 73-74: The sentence is unclear

Figure 1:

Figure 1B: What is the white box on the right? Is it supposed to be a scale indicator for color? Then color is missing

Figure 1D: The small numbers on the axis is impossible to read

In this and the other figures there are a lot of small boxes with question marks inside (some font problem seems to have occurred)

Fig. S1D: What is the blue peak in HCT116?

Figure 4:

Figure 4B: What is the white box on the right? Is it supposed to be a scale indicator for color? Then color is missing

Figure 4D-G: The authors should list all the genes that are significantly changes in a supp table (also for t0)

Fig. 5.

Figure 5A: What is the white box on the right? Is it supposed to be a scale indicator for color? Then color is missing

Fig. 5E: It is impossible to read the red text on the scale bar.

Fig. 5F: The legend for the y axis says mM (shouldn’t it be micrometer??)

Fig. 5E and F. The authors must include a control for the halo experiment. (ie a condition known to affect chr loop size) to show that the method is working.

Fig. 5G. It is not clear what is shown. The legend refers to ASY but I cannot see it in the panel. It is also not clear what A and D refers to.

Fig. S5E. It is not clear what is shown (what is the blue signal along the chromosome?)

The discussion could include a bit more on the change in cell line clustering after APH shown in fig. 4 H and I. What could underlie this change?

In most of the figure legends it is not stated what the error bars in the figures represent

Throughout the manuscript ChiP should be changed to ChIP

Author Response

Reviewer 2

Comments and Suggestions for Authors

In the manuscript “Low replicative stress triggers cell-type specific inheritable advanced replication timing” Courtot et al present data describing changes in replication timing after replication stress. The work is based on genome wide analysis of replication timing and the correlation to other genomic features. The work shows that replication timing is differentially affected after replication stress in the six different cell lines analysed. Moreover, the work finds that certain large regions in two of the cell lines show advanced replication timing after replication stress.

This work is purely descriptive and suffers from a lack of mechanistic insight. Nevertheless, it addresses a highly interesting subject, most of the experiments are well performed, and the data is clearly presented in the manuscript.

Major points.

  1. In Sup fig 1 there is no effect on CHK1-P of the APH treatment used in MRC5-N cells and RPE-1 cells (and minimal effect in U2OS and HCT116). Thus, the only cells that seems to have a clear response are also those with highest percent of genome affected. Therefore, the difference between the cell lines in this regard (Fig. 1B and may Fig 1C) is probably due to different levels of replication stress.

The authors cannot state in line 98 that replication stress is validated by Chk1 phosphorylation and they should in general be careful not to push their argumentation about cell line differences in inherited RT effects too much.

We changed the sentence by: “Aphidicolin treatment induces a low level of Chk1 phosphorylation on serine 345 compared to acute HU treatment and we noticed that MRC5-N and RPE -1 cells have the lowest P-Ckk1 induction (Figure S1A,B)”

We also add the following sentences in discussion L373-374 and 427-428

In Fig. 3 how are CFSs defined? It is not clear. It is well established that CFSs are cell type specific, therefore this must be taken into consideration for each of the six cell lines. The authors should correlate ADV and DEL to giant genes (for instance larger than 1 megabase) that are known to be expressed in the specific cell line.

This would provide a much more specific analysis of CFSs replication timing after replication stress. This could give important insight given the well-established effects of APH induced replication stress on CFSs.

For our study analyzing the overlap between aRTIL and CFS we didn’t focus on CFS that are specifically affected in each cell type. The list published in Brison O, et al. Nature com correspond to CFS mapped by conventional cytogenetics in several cell types. One of the reasons was that CFS are not defined in RKO cells.

As suggested, we added large gene coverage (>400kb) quantification in aRTIL that we previously have done (new fig S5B), and description in the manuscript L…. This additional analysis show that large genes are exclusively associated to DEL aRTIL in HCT116, MRC5-N and RPE-1 cells, that also have high level of DEL aRTIL and DEL aRTIL identified within CFS genomic regions. Large genes also have been identified in ADV aRTIL from RKO, K562 and U20S cells. This additional analysis suggest the association of DEL aRTIL with CFS and confirm that CFS can harbour RT advances in RKO and K562 cell lines.

Minor points.

Introduction line 46 and 48: The A and B compartment should be introduced better.

The sentence has been modified: line 45-49.

Introduction line 73-74: The sentence is unclear

The sentence has been simplified: L73-74

Concerning the following comments, we really apologize for all the problems of conversion of the files on the quality of the figures, the absence of colors on certain parts and typology problems. We worked to resolved all of them.

Figure 1:

Figure 1B: What is the white box on the right? Is it supposed to be a scale indicator for color? Then color is missing

Thank you, hope that it is resolved.

Figure 1D: The small numbers on the axis is impossible to read

This has been changed.

In this and the other figures there are a lot of small boxes with question marks inside (some font problem seems to have occurred)

Thank you, hope that it is resolved.

Fig. S1D: What is the blue peak in HCT116?

In fact, purple correspond to overlap of pink (APH) and blue (DMSO). Blue appear only when there is no overlap between APH and DMSO facs profiles. Labels has been modified on the figure

Figure 4:

Figure 4B: What is the white box on the right? Is it supposed to be a scale indicator for color? Then color is missing

It has been changed.

Figure 4D-G: The authors should list all the genes that are significantly changes in a supp table (also for t0)

The list is now presented in Figure 4D and F.

Fig. 5.

Figure 5A: What is the white box on the right? Is it supposed to be a scale indicator for color? Then color is missing

It has been changed

Fig. 5E: It is impossible to read the red text on the scale bar.

Scale bar value is now indicated in the legend.

Fig. 5F: The legend for the y axis says mM (shouldn’t it be micrometer??)

Thank you, this mistake has been corrected.

Fig. 5E and F. The authors must include a control for the halo experiment. (ie a condition known to affect chr loop size) to show that the method is working.

We did not include a control because for us the DMSO was the reference and the fact that we did not see any differences between the two conditions in the RPE-1 cells (Figure S8 E-F) was a good confirmation that the technique is working.

Fig. 5G. It is not clear what is shown. The legend refers to ASY but I cannot see it in the panel. It is also not clear what A and D refers to.

The “A” was for APH and “D” for DMSO, we changed the nomenclature on the figure and the legend has been corrected.

Fig. S5E. It is not clear what is shown (what is the blue signal along the chromosome?)

The blue signal is the normal RT, it is now better explained in the legend.

By this illustration, we wanted to show the normal timing of the impacted domains and also that there are more intersections between the ADV RKO and K562 and more between DELs of the others cell lines in complement to Figure 2B

The discussion could include a bit more on the change in cell line clustering after APH shown in fig. 4 H and I. What could underlie this change?

As RKO and HCT116 are two colon cancer cell lines, it was expected to see them clustering in the DMSO condition. The fact that they are further apart in the APH condition reveals that the strong impact of the drug in RKO affect the identity of the cells, as they do not anymore aggregate with the other colon cancer line HCT116, in which RT modifications are much lower.

In most of the figure legends it is not stated what the error bars in the figures represent.

Thank you for noticing, this has been corrected.

Throughout the manuscript ChiP should be changed to ChIP

Thank you, it has been corrected.

Reviewer 3 Report

The manuscript submitted by Courtot et al., entitled, “Low replicative stress triggers cell-type specific inheritable advanced replication timing”  that primarily focusses on DNA replication timing in response to aphidicolin. It’s an interesting study but needs following corrections to get it published.

Major correction:

  1. All the symbols needs to be corrected in the manuscript.
  2. Immunofluorescence imaging of H2AX in response to aphidicolin treatment can be a great supporting experiment to prove the fact that RT delays are associated with DNA damages and RT advances are protective against DNA damages (Figure 3).
  3. A curious question: What will happen if you treat the cells continuously over 3-4 generations with amphidicolin and then release the cells to see whether RT modifications within aRTIL are reversible or not?

Minor corrections:

  1. Several errors in labelling of Figure 1, There is “?” sign at several places. In figure 1B the labelling of cell line is off like HCT116 cells.
  2. Labelling needs to be diligently checked throughout the figures, specially the places with symbols.

Author Response

Open Review 3

Comments and Suggestions for Authors

The manuscript submitted by Courtot et al., entitled, “Low replicative stress triggers cell-type specific inheritable advanced replication timing” that primarily focusses on DNA replication timing in response to aphidicolin. It’s an interesting study but needs following corrections to get it published.

Major correction:

All the symbols needs to be corrected in the manuscript.

Thank you for this comment, we paid attention to correct all the symbols.

Immunofluorescence imaging of H2AX in response to aphidicolin treatment can be a great supporting experiment to prove the fact that RT delays are associated with DNA damages and RT advances are protective against DNA damages (Figure 3).

Unfortunately, by immunofluorescence imaging of H2AX, no specific genomic region can be attributed to the detected signal (foci). We cannot distinguish delayed from advanced domains targeted by H2AX by this approach. The best one is ChIP-seq, that’s why we performed the analyses on this type of data.

A curious question: What will happen if you treat the cells continuously over 3-4 generations with amphidicolin and then release the cells to see whether RT modifications within aRTIL are reversible or not?

 This an interesting point that we discussed (L404-410), unfortunately, we do not have the answer.

Minor corrections:

We really apologize for all the problems of conversion of the files on the quality of the figures, the absence of colors on certain parts and typology problems. We worked to resolved all of them.

Several errors in labelling of Figure 1, There is “?” sign at several places. In figure 1B the labelling of cell line is off like HCT116 cells.

Labelling needs to be diligently checked throughout the figures, specially the places with symbols.

Round 2

Reviewer 1 Report

Dear Editor

I decided to recommend rejecting the revised manuscript in spite of the fact that the authors made some improvements.

Some of the main issues that remain are -

  1. The main finding is the uniqueness of RKO cells yet the message of the paper remains the diversity between cells in response to mild stress, which is true regarding the unique response of RKO but I am still not convinced about the response of the other cells.
  2. In the authors’ response to the lack of overlap with CFS they suggest that their measurements are not accurate for extremely delayed RT (that occur in G2). I had not thought of this, but once suggested it might explain the advanced RT in RKO, since it may be actually delayed to G2 and therefore captured by their assay as middle, these late regions become now middle and thus captured as advanced. This point should be sorted out.
  3. In order to account for my request to show convincing evidence about the accuracy of differential regions in other cell lines, the authors add two figures. These types of questions can be answered only with statistics and not with a few examples that may not be representative.
  4. I have raised the question of the length of G2 and I was not convinced by the authors’ response since a one-hour labeling in the late S will result with G1 labeling after 16 hours, thus S length is not relevant, only the G2 length.
  5. The Holm method for multiple hypothesis correction is indeed valid, however the question is how many hypotheses they corrected for, information that they do not provide.
  6. The Jaccard index is indeed a metric but without repeating the permutation 1000 times and comparing the actual overlap with the distribution of the permutations, As done by Kim et al (the paper they cite in their response) it has no statistical significance.

Author Response

2nd Review from Reviewer 1 and Academic Editors’ decision and Coments

2nd review from reviewer 1 – Negative

Comments:

I decided to recommend rejecting the revised manuscript in spite of the fact that the authors made some improvements.

Some of the main issues that remain are -

  1. The main finding is the uniqueness of RKO cells yet the message of the paper remains the diversity between cells in response to mild stress, which is true regarding the unique response of RKO but I am still not convinced about the response of the other cells.

We provided reproducible independent experiments clearly showing evidence about the strength of the data. Moreover, the experiments were performed under the same conditions experimental for RKO than other cells.

  1. In the authors’ response to the lack of overlap with CFS they suggest that their measurements are not accurate for extremely delayed RT (that occur in G2). I had not thought of this, but once suggested it might explain the advanced RT in RKO, since it may be actually delayed to G2 and therefore captured by their assay as middle, these late regions become now middle and thus captured as advanced. This point should be sorted out.

According to our methodology, replication timing is considered to be early when the curve representing RT profile is above zero, late when the curve goes below zero and middle when the curve is at the zero. In our case, the RT profile of advanced regions in the APH condition is never at the level of zero but well above for the RKO and below for the K562.This indicates that late regions that are advanced in response to APH do not become middle but early in RKO and remain late for K562 (figure 1 and S2) and therefore that the advanced regions we have detected cannot be a technical artefact or “delayed to G2”.

Moreover, reviewer 1 assumes that the ADV aRTIL are actually delays that we do not detect since we do not take the entire G2 phase. We are not in this type of configuration since we detect neosynthesized DNA from the S1 fraction that hybridizes on the chip probes corresponding to these advanced regions indicating that material is present and not absent as assumed by reviewer 1.

Importantly, we also validate ADV aRTIL from RKO by, an independent technology, qPCR. As for microarrays, we showed that there is an enrichment of the DNA amount within these regions in the s1 fraction in the Aphidicolin condition (APH) compared to DMSO confirming that the RT is indeed advanced and not delayed within these regions (figure S3).

  1. In order to account for my request to show convincing evidence about the accuracy of differential regions in other cell lines, the authors add two figures. These types of questions can be answered only with statistics and not with a few examples that may not be representative.

We have proved in figures S2 and S6 as well as in the 2 figures added in the first answer to reviewer 1 the validity of our experiments. We used START- R to have the significant differences between our control (DMSO) and aphidicolin treated condition on the 2 replicates.

  1. I have raised the question of the length of G2 and I was not convinced by the authors’ response since a one-hour labeling in the late S will result with G1 labeling after 16 hours, thus S length is not relevant, only the G2 length.

We didn’t claim that we don’t have a longer G2 as we saw a mild increase of G2/M cell number. However, we mainly observed an accumulation of treated cells in S phase in all cell lines (Figure S1C, D, E). In RKO, it is likely that cells detected in G1 phase after 16 hours of treatment were at the end of S-phase when the treatment started and they reached the end of G1 at the end of the treatment. Since the G1 phase duration is between 8 and 10 hours and mitosis about 1 hour, this suggests that these cells spent a maximum of 5 to 7 hours (and not almost 16h) in G2 versus about 4 to 5 hours without treatment. Cells that were more inside S-phase when APH treatment started were more affected than very late S-phase ones and more retained into S-phase. In any case, this point does not affect the fact that the replication timing analyzed at T0 is performed on cells that have undergone aphidicolin treatment on a single S phase (L 97-103).

  1. The Holm method for multiple hypothesis correction is indeed valid, however the question is how many hypotheses they corrected for, information that they do not provide.

We will try to explain in more detail to the reviewer how the differential analysis has been done with Start-R, it has been also described in the publication in NAR G&B (https://academic.oup.com/nargab/article/2/2/lqaa045/5859925?searchresult=1);

START-R uses the p.adjust function of R which is defined as is by this documentation: https://www.rdocumentation.org/packages/stats/versions/3.6.2/topics/p.adjust

In this function, the parameter n is left by default as it is recommended by the R documentation. This number n depends on the number of windows to examine two by two between the two experiments to compare. This number depends on the size of the windows, the size of the overlap between each window and the coverage of the genome studied. Here we have left the default parameters of window size = 60 bp and overlap size = 30. So we have about 60 000 000 windows per experiment to compare two by two (between each experiment) by a T-Test which will generate a pValue. This last one will be adjusted by the p.adjust function with the Holm method as explained and validated before in the publication in NAR G&B

If the reviewer wants to know more, I invite him to look at the lines of code of START-R between the line 3234 and 3251 available on the Github site: https://github.com/thomasdenecker/START-R

I sincerely hope that our explanation has been clearer the reviewer 1 this time.

  1. The Jaccard index is indeed a metric but without repeating the permutation 1000 times and comparing the actual overlap with the distribution of the permutations, As done by Kim et al (the paper they cite in their response) it has no statistical significance.

We performed the Jaccard index for the overlap between CFS and observed aRTIL or between CFS and 1000 instances of randomized intervals. Then we measured the statistical significance by calculating the z-score (values added on Figures 2E, 3B and S5A). We did the same type of analysis for the overlaps of DEL and ADV aRTIL from each cell lines (Figure 2 E). We implemented the material and method section 4.6 and the legend of the figures to better describe this analysis procedure and we hope that it is now clearer and convincing.

Academic edfitors’ decision – Accept after minor revisions

Comments:

Reviewer 1 raised several issues and does not accept the paper at this stage. However reviewers 2 and 3 are accepting the manuscript.

At this step we would like the authors to reply to the 6 comments that he addressed :

Especially it would be important to answer to points 2 and 4 ; reply to 5 and 6 are less important but still interesting ; 1 and 3 are negligible.

We are looking forward to your answers.

We sincerely want to thanks the additional edditors for taking the time to read the paper, reviews and to give us the opportunity to answer to the second review from reviewer 1.
